

# Gravity disturbance driven ocean circulation

Peter C. Chu[1n]

[1]Department of Oceanography, Naval Postgraduate School, Monterey, CA 93943, USA

*Correspondence to*: Peter C. Chu (pcchu@nps.edu)

**Abstract.** The Earth true gravity ($\mathbf{g}$) has been simplified in oceanography and meteorology into the standard gravity $\mathbf{g}_s$ (= -$g_0\mathbf{k}$, $g_0$ = 9.81 m s$^{-2}$) with $\mathbf{k}$ the unit vector perpendicular to the spherical surface or the normal $\mathbf{g}_n$ [= -$g(\varphi)\mathbf{K}$] with $\mathbf{K}$ the unit vector perpendicular to the ellipsoidal surface. The gravity disturbance ($\delta\mathbf{g} = \mathbf{g} - \mathbf{g}_n$) due to nonuniform Earth mass density is totally neglected.  After including the gravity disturbance into the combined Sverdrup-Stommel-Munk equation for ocean circulation, the volume transport stream-function ($\Psi$) is driven by both gravity disturbance forcing (GDF) and surface wind

forcing (i.e., curl $\boldsymbol{\tau}$) with $\boldsymbol{\tau}$ the surface wind stress.  The non-dimensional $F$ number (i.e., ratio of global |GDF| versus global |curl $\boldsymbol{\tau}$|) is estimated as 0.6918 using three publicly available datasets in climatological, geodetic, and oceanographic communities.  Such an $F$-value (0.6918) clearly shows the comparable GDF and surface wind stress curl in driving ocean circulation, and the urgency to include the gravity disturbance in ocean dynamics. Besides, this study also cleared up some misconceptions in gravity related valuables such as vertical, geopotential, marine geoid, and dynamic ocean topography.

**1 Introduction**

    The Earth gravity and associated variables such as vertical, geopotential, geoid, and dynamic ocean topography have three types: standard gravity ($\mathbf{g}_s$), normal gravity [$\mathbf{g}_n$, called effective gravity in Vallis (2017), or apparent gravity in Bernard (2015)] and true gravity ($\mathbf{g}$) [called real gravity in Chu (2021a)].  The standard gravity $\mathbf{g}_s$ (= -$g_0\mathbf{k}$, $g_0$ = 9.81 m s$^{-2}$) is for the Earth being assumed as a rigid sphere with uniform mass density and without rotation; $\mathbf{k}$ is the unit vector representing the standard-vertical

and perpendicular to the spherical surface (upward positive).  The normal gravity $\mathbf{g}_n$ [= -$g(\varphi)\mathbf{K}$] is for this rigid spherical Earth with rotation (becoming uniform ellipsoid) and $\mathbf{K}$ the unit vector representing the normal-vertical and perpendicular to the ellipsoidal surface (upward positive); $\varphi$ is the latitude. The true gravity

$$\mathbf{g}(\lambda, \varphi, z) = \mathbf{g}_n + \delta\mathbf{g}(\lambda, \varphi, z) \tag{1}$$

    is for Earth with rotation and nonuniform mass density and represents the true-vertical. Here, $\delta\mathbf{g}$ is the gravity disturbance

(with $\lambda$ the longitude and $z$ the depth), and independent on Earth rotation (Hackney and Featherstone 2003).  Obviously, $\mathbf{g}_s$ ($\mathbf{g}_n$) doesn't have longitudinal-latitudinal component in spherical (oblate spheroid) coordinates.

    Geopotential in oceanography and meteorology is the same as the gravity potential in geodesy but with opposite signs. Geopotential also has three types with corresponding gravity: standard-geopotential ($\Phi_s = g_0 z$) with $\mathbf{g}_s$, normal-geopotential [$\Phi_n = g(\varphi)z$] with $\mathbf{g}_n$, and true-geopotential

$\Phi = \Phi_n - T \tag{2a}$





with **g**. $T(\lambda, \varphi, z)$ is the disturbing gravity potential (Kostelecký et al. 2015)

$$T(\lambda,\varphi,z) = \frac{GM}{(R+z)}\sum_{l=2}^{\infty}\sum_{m=0}^{l}\left(\frac{R}{R+z}\right)^{l}\left[\left(C_{l,m} - C_{l,m}^{el}\right)\cos m\lambda + S_{l,m}\sin m\lambda\right]P_{l,m}(\sin\varphi) \qquad (2b)$$

which is quantified by gravity models along with observations in geodetic community. Here, $G$ ( $6.674\times10^{-11}\text{m}^3\text{kg}^{-1}\text{s}^{-2}$) is the

gravitational constant; $M$ ($5.9736 \times 10^{24}$ kg) is the mass of the Earth; $R$ ($6.3781364\times10^6$ m) is the Earth radius; ( $C_{l,m}, C_{l,m}^{el}, S_{l,m}$

) are the harmonic geopotential coefficients with $C_{l,m}^{el}$ belonging to the reference ellipsoid;   and   $P_{l,m}(\sin\varphi)$ are the Legendre

associated functions with $(l, m)$ the degree and order of the harmonic expansion.

Since **K** is deviated from **k**, the normal gravity ($\mathbf{g}_n$) is represented in the spherical coordinate

$$\mathbf{g}_n = \mathbf{g}_n^{(h)} - g(\varphi)(\mathbf{K}\bullet\mathbf{k})\mathbf{k}, \quad \mathbf{g}_n^{(h)} = [-g(\varphi)][(\mathbf{K}\bullet\mathbf{i})\mathbf{i} + (\mathbf{K}\bullet\mathbf{j})\mathbf{j}] \qquad (3)$$

where $\mathbf{g}_n^{(h)}$ is longitudinal-latitudinal component of $\mathbf{g}_n$ in the polar spherical coordinates, which is much smaller than $\delta\mathbf{g}$ [see

any gravity model (e.g., Sandwell and Smith 1997) and geodesy text book (e.g., Vaniček and Krakiwsky (1986)],

$$O(|\mathbf{g}_n^{(h)}|) << O(|\delta\mathbf{g}|) \qquad (4)$$

The standard-geopotential (normal-geopotential) surfaces are spherical (ellipsoidal) surfaces whose top level for the ocean is

represented by

$$r = R, \text{ or } z = 0 \text{ (local coordinate)} \qquad (5)$$

which are non-undulated marine standard-geoid surface ($N_s$) and normal-geoid surface ($N_n$),

$$\nabla N_s = 0, \quad \nabla^{(obl)}N_n = 0 \qquad (6)$$

where   $\nabla \equiv \mathbf{i}\partial / (R\cos\varphi\partial\lambda) + \mathbf{j}\partial / (R\partial\varphi)$   is the two-dimensional vector differential operator in the polar spherical coordinates

with $(\mathbf{i}, \mathbf{j})$ the corresponding longitudinal-latitudinal unit vectors; and   $\nabla^{(obl)}$ is for the oblate spheroid coordinates. Both marine

standard-geoid and normal-geoid surfaces are evidently different from the marine true-geoid surface ($N$),

$r = R + N(\lambda,\varphi),$ or  $z = N(\lambda,\varphi)$ (local coordinate)      (7)

which is undulated.

Meteorological and oceanographic communities are aware of the differences between $\mathbf{g}_n$ and $\mathbf{g}_s$, $\Phi_n$ and $\Phi_s$, as well as

associated oblate spheroid and polar spherical coordinates  (e.g., Haurwitz 1941; Philips 1973; Gill 1982; Holton 2004; Gates

2004; Pedlosky 2013; Benard 2015;  Vallis 2017), and have reached the consensus on  replacement of the normal gravity by

the standard gravity   $(\mathbf{g}_n \approx \mathbf{g}_s)$  and on use of the polar spherical coordinates $(\lambda,\ \varphi,\ z)$   instead of the oblate spheroid

coordinates with very small geometric error about 0.17% (Gill 1982) or 0.3% (Beńard 2015), although the errors may be

accumulated after long-term integration (Gates 2004;  Beńard 2015). In addition to the polar spherical coordinates $(\lambda,\ \varphi,\ z)$,

meteorological and oceanographic communities also use the local coordinates $(x, y, z)$,


$$\frac{\partial}{\partial x} = \frac{1}{R\cos\varphi}\frac{\partial}{\partial\lambda}, \quad \frac{\partial}{\partial y} = \frac{1}{R}\frac{\partial}{\partial\varphi} \tag{8}$$

where the $(x, y)$ plane is perpendicular to $\mathbf{k}$. Let $S$ be the sea surface. With the standard gravity $(-g_0\mathbf{k})$, the dynamic ocean topography is defined by $D = S - N_s$. Since the marine standard-geoid surface is non-undulated, we have

$$\nabla D = \nabla(S - N_s) = \nabla S \tag{9}$$

However, both meteorological and oceanographic communities never use the true gravity $\mathbf{g}$, i.e., never include the gravity disturbance $(T)$. Following the meteorological and oceanographic communities' assessment on $\mathbf{g}_n$ ($\mathbf{g}_n \approx \mathbf{g}_s$), Chu (2021a)

suggested an approximate true-geopotential for meteorology and oceanography

$$\Phi = \Phi_S - T, \quad \Phi_S = g_0 z, \quad T(\lambda,\varphi,z) \approx T(\lambda,\varphi,0) = g_0 N(\lambda,\varphi) \tag{10}$$

and showed the importance of the gravity disturbance $(T)$ on atmospheric and oceanic Ekman layer dynamics (Chu 2021b, c).

With the true gravity $(\mathbf{g})$, the dynamic ocean topography satisfies

$$\nabla D = \nabla(S - N) \neq \nabla S \tag{11}$$

which is totally different from that with the standard gravity $\mathbf{g}_s$, i.e., Eq(9). Table 1 summarize the difference among the three types of the gravity and associated variables.

To show further evidence of importance of the true gravity $\mathbf{g}$ in ocean dynamics, the gravity disturbance $(T)$ is included in the theoretical Sverdrup-Stommel-Munk equation in addition to the surface wind forcing. The ocean circulation is driven by both gravity disturbance and wind. Section 2 presents the basic dynamic equation. Section 3 describes combined Sverdrup-

Stommel-Munk equation with the gravity disturbance. Section 4 shows the publicly available data sources. Section 5 presents the comparable global gravity disturbance forcing (GDF) and surface wind stress curl computed from the data depicted in Section 4. Section 6 demonstrates the global Sverdrup and Stommel volume transports driven by comparable GDF and surface wind stress curl. Section 7 presents the conclusions. Appendix A presents the derivation of the combined Sverdup-Stommel-Munk equation including the gravity disturbance.

**2 Basic equations**

Large-scale ocean circulation under the Boussinesq approximation is governed by the momentum equation (Chu, 2021a)

$$\rho_0\left[\frac{D\mathbf{U}}{Dt} + f\mathbf{k}\times\mathbf{U}\right] = -\nabla_3 p + \rho\mathbf{g} + \rho_0(\mathbf{F}_h + \mathbf{F}_v) \tag{12}$$

and the continuity equation

$$\nabla\bullet\mathbf{U} + \frac{\partial w}{\partial z} = 0 \tag{13}$$





where $\nabla_3 \equiv \nabla + \mathbf{k}\partial / \partial z$ , is the three-dimensional vector differential operator; $D/Dt$ is total time derivative; $\mathbf{U} = (u, v)$, is the

2D velocity vector in $(\mathbf{i}, \mathbf{j})$ surfaces; $w$ is the z-component velocity; $p$ is the pressure; $\rho$ is the sea water density; $\rho_0 = 1{,}028$ kg

m$^{-3}$; $f = 2\Omega \sin \varphi$, is the Coriolis parameter with $\Omega = 2\pi/(86164 \text{ s})$; $(\mathbf{F}_h, \mathbf{F}_v)$ are the frictional forces with lateral and z-directional

shears represented by

$$\mathbf{F}_h = A\nabla^2 \mathbf{U}, \quad \mathbf{F}_v = \frac{\partial}{\partial z}\left( K\frac{\partial \mathbf{U}}{\partial z} \right) \tag{14}$$

where $(A, K)$ are the corresponding eddy viscosities.

With the constant reference density $\rho_0$, three-dimensional hydrostatic equilibrium between the pressure gradient force and the

true gravity $\mathbf{g}$ $[= \nabla_3 \Phi, \quad \Phi = g_0 z - T \approx g_0 z - g_0 N(\lambda, \varphi) ]$ is given by

$$-\nabla_3 p_0 - \rho_0 \nabla_3 \Phi = 0 \tag{15}$$

where

$$p_0 = -\rho_0 g_0 z + \rho_0 T \approx -\rho_0 g_0 (z - N) \tag{16}$$

is the hydrostatic pressure. Subtraction of (15) from (12) leads to

$$\rho_0 \left[ \frac{D\mathbf{U}}{Dt} + f\mathbf{k} \times \mathbf{U} \right] = -\nabla \hat{p} + (\rho - \rho_0)\nabla T + \rho_0 (\mathbf{F}_h + \mathbf{F}_v) \tag{17a}$$

$$\frac{\partial \hat{p}}{\partial z} = -(\rho - \rho_0)g_0 \tag{17b}$$

where   $\hat{p} = p - p_0$, is the dynamic pressure.

**3  Combined Sverdrup-Stommel-Munk equation**

For steady-state low Rossby number (negligible nonlinear advection) flow with friction (i.e., $D\mathbf{U}/Dt = 0$, and $\mathbf{F}_h \neq 0$, $\mathbf{F}_v \neq$

0), Eq.(17a) is simplified into

$$f\mathbf{k} \times \mathbf{U} = -\frac{1}{\rho_0}\nabla \hat{p} + \frac{\rho - \rho_0}{\rho_0}\nabla T + (\mathbf{F}_h + \mathbf{F}_v) . \tag{18}$$

With the wind stress, $\boldsymbol{\tau} = (\tau_x, \tau_y)$, as the forcing at the rigid-lid ocean surface ($z = 0$) and negligible lower boundary ($z = -H$)

stress (Sverdrup 1947, Munk 1950) or taking the Rayleigh friction at the lower boundary (Stommel 1948), a combined

Sverdrup-Stommel-Munk equation in the local coordinate system (8) as the tradition is derived from (18) (see derivation in

Appendix A)



$$-A\nabla^4\Psi + \gamma\nabla^2\Psi + \beta\frac{\partial\Psi}{\partial x} = \frac{1}{\rho_0}\left[\text{curl }\boldsymbol{\tau} + \int_{-H}^{0} J(\rho,T)dz\right], \quad J(\rho,T) \equiv \frac{\partial\rho}{\partial x}\frac{\partial T}{\partial y} - \frac{\partial T}{\partial x}\frac{\partial\rho}{\partial y}, \quad T \approx g_0 N \qquad (19)$$

where $\beta = (2\Omega\cos\varphi)/R$; $J(\rho, T)$ is the Jacobian of $\rho$ and $T$; and $\Psi$ is the volume transport stream-function defined by

$$\frac{\partial\Psi}{\partial x} = \int_{-H}^{0} vdz, \qquad \frac{\partial\Psi}{\partial y} = -\int_{-H}^{0} udz \qquad (20)$$

After changing the flat lower boundary into non-flat bottom topography, $z = -H(x, y)$, Eq.(19) becomes,

$$-A\nabla^4\Psi + \gamma\nabla^2\Psi + \beta\frac{\partial\Psi}{\partial x} = \frac{1}{\rho_0}\left[\text{curl }\boldsymbol{\tau} + \int_{-H(x,y)}^{0} J(\rho,T)dz\right] + \begin{array}{l}\text{Bottom Topographic}\\ \text{Effect Term}\end{array} \qquad (21)$$

Note that the bottom topographic effect on the volume transport is beyond the scope of this study, and therefore is not identified. The second term in the righthand side is an additional term called the gravity disturbance forcing (GDF)

$$\text{GDF} = \int_{-H(x,y)}^{0}\left(\frac{\partial\rho}{\partial x}\frac{\partial T}{\partial y} - \frac{\partial\rho}{\partial y}\frac{\partial T}{\partial x}\right)dz, \quad T \approx g_0 N \qquad (22)$$

A non-dimensional $F$ number is defined by

$$F = \frac{O\left[\left|\text{GDF}(\lambda,\varphi)\right|\right]}{O\left[\left|\text{curl }\boldsymbol{\tau}(\lambda,\varphi)\right|\right]} \qquad (23)$$

to identify the importance of GDF versus the surface wind stress curl.

**4 Data sources**

Three publicly available datasets were used in this study: (a) the global static gravity model EIGEN-6C4 (Förste, et al., 2014; Kostelecký et al., 2015) for the geoid undulation $N(\lambda, \varphi)$ (Figure 1a), (b) the climatological annual mean temperature and salinity from the NCEI WOA18 (Boyer et al., 2018) for the sea water density $\rho(\lambda, \varphi, z)$, and (c) the climatological annual mean surface wind stress $(\tau_\lambda, \tau_\varphi)$ (da Silva et al., 1994). These datasets are used to identify the importance of GDF versus wind stress curl.

**5 Global GDF and surface wind stress curl**

The GDF is calculated by Eq.(22) using the density $\rho(x, y, z)$ from the WOA18 annual mean temperature and salinity data and the true-geoid undulation $N(x, y)$ from the EIGEN-6C4 data. The surface wind stress curl is computed from the SMD94 annual mean surface wind stress $(\tau_x, \tau_y)$ data. The calculated global GDF (simplified as 'J' in Figure 1b, c) and surface wind stress curl (Figure 2) have comparable magnitudes with different horizontal distributions (Figure 1b and Figure 2a). The histograms

of |GDF| (Figure 1c) and (|curl $\boldsymbol{\tau}$|) (Figure 2b) show near Gamma distribution. |GDF| has comparable mean and standard deviation $(3.448, 4.283)\times10^{-8}\text{Nm}^{-3}$, with |curl $\boldsymbol{\tau}$| $(4.984, 4.052)\times10^{-8}\text{Nm}^{-3}$; but has two-time larger skewness and kurtosis (2.19,





8.12), than |curl $\boldsymbol{\tau}$| (1.081, 4.137). If the global mean value is treated as the order of magnitude, the $F$-number defined by (23) can be estimated by

$$F = \frac{Mean\left[\left|\text{GDF}(\lambda, \varphi)\right|\right]}{Mean\left[\left|\text{curl } \boldsymbol{\tau}(\lambda, \varphi)\right|\right]} = \frac{3.448 \times 10^{-8} \text{ N m}^{-3}}{4.984 \times 10^{-8} \text{ N m}^{-3}} = 0.6918 \tag{24}$$

which shows the two forcing functions (GDF and wind stress curl) are comparable to drive the ocean circulation. Note that large |GDF| values occurring around the Gulf Stream and Antarctic Circumpolar Circulation regions. The reason is explained as follows. From Eq.(22) the GDF can be rewritten by

$$\text{GDF} = \mathbf{k} \bullet \left(\mathbf{B} \times \nabla T\right) = g_0 \left|\mathbf{B}\right|\left|\nabla N\right| \sin \alpha, \quad \mathbf{B} \equiv \int_{-H}^{0} \nabla \rho \, dz \tag{25}$$

where the vector $\mathbf{B}$ represents the baroclinicity; and $\alpha$ is the angle between $\mathbf{B}$ and $\nabla N$ . The |GDF| value depends on the angle

$\alpha$ and the intensities of the two vectors $|\mathbf{B}|$ and $|\nabla N|$. Near the Gulf Stream and Antarctic Circumpolar Circulation regions, the vector $\mathbf{B}$ is in the north-south direction usually with large magnitude. However, $\nabla N$ is in the east-west direction (Figure 1a) with noticeable magnitude (i.e., $|\nabla N|$). Near 90° cross angle $\alpha$ may be the major reason to cause large |GDF| values there.

**6 Sverdrup and Stommel volume transports**

The Sverdrup equation is obtained by setting A = 0 (no deflected-horizontal eddy viscosity), and $\gamma$ = 0 (no bottom friction) in

145    (19)

$$\beta \frac{\partial \Psi}{\partial x} = \frac{1}{\rho_0} \left[ \text{curl } \boldsymbol{\tau} + g_0 \int_{-H}^{0} J(\rho, N) dz \right] \tag{26}$$

The Stommel equation is obtained by setting A = 0 (no deflected-horizontal eddy viscosity) in (19),

$$\gamma \nabla^2 \Psi + \beta \frac{\partial \Psi}{\partial x} = \frac{1}{\rho_0} \left[ \text{curl } \boldsymbol{\tau} + g_0 \int_{-H}^{0} J(\rho, N) dz \right], \quad \gamma = 10^{-6} \tag{27}$$

The standard boundary conditions are used: $\Psi$ = 0 at the eastern boundary for the Sverdrup equation, and at all boundaries for

the Stommel equation. In the Southern Ocean, the cyclic boundary condition is used at 20°E section across the Africa and Antarctic continents. Three numerical integrations are conducted to solve Eqs(26) and (27) for the Sverdrup (Figure 3) and Stommel (Figure 4) volume transport stream-functions: (a) $\Psi$ with both (curl $\boldsymbol{\tau}$) and GDF, (b) $\Psi_1$ with (curl $\boldsymbol{\tau}$) only, and (c) $\Psi_2$ with GDF. Relative root mean differences between ($\Psi$, $\Psi_1$) and ($\Psi$, $\Psi_2$) for both equations,

$$\text{RRMSD}(\Psi, \Psi_k) = \frac{\sqrt{\sum_i \sum_j \left[\Psi(i, j) - \Psi_k(i, j)\right]^2}}{\sqrt{\sum_i \sum_j \left[\Psi(i, j)\right]^2}}, \quad k = 1, 2 \tag{28}$$





are calculated with both GDF and winds ($\Psi$) taken as the reference. RRMSD between $\Psi$ (GDF and winds) (Figures 3a and 4a), $\Psi_1$ (winds only) (Figures 3b and 4b) is 0.373 for the Sverdrup volume transport stream-function, and 0.405 for the Stommel volume transport-stream function. RRMSD between $\Psi$ (GDF and winds) and $\Psi_2$ (GDF only) (Figures 3c and 4c) is 0.767 for the Sverdrup volume-transport stream-function, and 0.848 for the Stommel volume transport stream function. Thus, GDF cannot be neglected against the surface wind stress curl in ocean circulation.

**7 Conclusions**

Importance of the gravity disturbance ($T$) in ocean dynamics is demonstrated here with updating the combined Svedrup-Stommel-Munk equation [through replacement of the standard gravity (-$g_0\mathbf{k}$) by the true gravity $\mathbf{g}$] into a new equation with two forcing functions: gravity disturbance forcing (GDF) and surface wind stress curl. Both forcing functions have comparable magnitudes with the $F$-number (global |GDF| versus global |curl $\boldsymbol{\tau}$|) of 0.6918 using three independent and publicly available

global datasets: SMD94 for wind stress ($\boldsymbol{\tau}$), WOA18 for water density ($\rho$), and EIGEN-6C4 for the geoid ($N$). The relative difference in the volume transport (with using the true gravity as the reference) is evident between standard and true gravities such as 0.373 in the Sverdrup volume transport stream-function and 0.405 in the Stommel volume transport stream-function. Thus, the gravity disturbance should be included in ocean dynamics.

**Appendix A Combined Sverdrup-Stommel-Munk equation**

The momentum equation (18) is represented into component form,

$$-f\rho_0 v = -\frac{\partial \hat{p}}{\partial x} + (\rho - \rho_0)\frac{\partial T}{\partial x} + \rho_0 K \frac{\partial^2 u}{\partial z^2} + \rho_0 A\nabla^2 u ,\tag{A1}$$

$$f\rho_0 u = -\frac{\partial p}{\partial y} + (\rho - \rho_0)\frac{\partial T}{\partial y} + \rho_0 K \frac{\partial^2 v}{\partial z^2} + \rho_0 A\nabla^2 v ,\tag{A2}$$

The Sverdrup-Stommel-Munk theories assume rigid lid surface ($z = 0$) and flat bottom ($z = -H$),

$$w|_0 = 0, \quad w|_{-H} = 0\tag{A3}$$

and the wind stress ($\tau_x$, $\tau_y$) as the forcing at the ocean surface

$$\rho_0 K(\frac{\partial u}{\partial z}, \frac{\partial v}{\partial z})|_{z=0} = (\tau_x, \tau_y) .\tag{A4}$$

The bottom stress ($\tau_x^{(b)}$, $\tau_y^{(b)}$) was either neglected (Sverdrup, 1947; Munk, 1950), or taken as the Rayleigh friction (Stommel,

180    1948),

$$\rho_0 K(\frac{\partial u}{\partial z}, \frac{\partial v}{\partial z})|_{z=-H} = (\tau_x^{(b)}, \tau_y^{(b)}) = \gamma\rho_0(M_x, M_y),\tag{A5}$$





where

$$M_x = \int_{-H}^{0} u\,dz, \quad M_y = \int_{-H}^{0} v\,dz \tag{A6}$$

are the longitudinal and latitudinal volume transports per unit length, and $\gamma$ is the Rayleigh friction coefficient. Integration of

the continuity equation (13) with respect to $z$ and use of the boundary conditions (A3) leads to

$$\frac{\partial M_x}{\partial x} + \frac{\partial M_y}{\partial y} = 0 \tag{A7}$$

Here, the volume transport stream-function ($\Psi$) can be defined by

$$M_x = -\frac{\partial \Psi}{\partial y}, \quad M_y = \frac{\partial \Psi}{\partial x} \tag{A8}$$

Integration of the momentum equations (A1) and (A2) with respect to $z$ from $z = -H$ to $z = 0$ and use of (A4) and (A5) lead to

$$-A\nabla^2 M_x - fM_y = -\frac{1}{\rho_0} \int_{-H}^{0} \frac{\partial p}{\partial x} dz + \int_{-H}^{0} \frac{(\rho - \rho_0)}{\rho_0} \frac{\partial T}{\partial x} dz + \frac{\tau_x - \tau_x^{(b)}}{\rho_0} \tag{A9}$$

$$-A\nabla^2 M_y + fM_x = -\frac{1}{\rho_0} \int_{-H}^{0} \frac{\partial p}{\partial y} dz + \int_{-H}^{0} \frac{(\rho - \rho_0)}{\rho_0} \frac{\partial T}{\partial y} dz + \frac{\tau_y - \tau_y^{(b)}}{\rho_0} \tag{A10}$$

Cross differentiation of (A9) and (A10) with respect to $x$ and $y$ leads to the combined Sverdrup-Stommel-Munk equation with GDF, i.e., Eq.(19),

$$\boxed{-A\nabla^4 \Psi + \gamma \nabla^2 \Psi + \beta \frac{\partial \Psi}{\partial x} = \frac{1}{\rho_0}\left[\text{curl } \boldsymbol{\tau} + \int_{-H}^{0} J(\rho, T)dz\right], \quad J(\rho, T) \equiv \frac{\partial \rho}{\partial x}\frac{\partial T}{\partial y} - \frac{\partial \rho}{\partial y}\frac{\partial T}{\partial x}} \tag{A11}$$

where $\beta = (2\Omega\cos\varphi)/R$. Without the gravity disturbance (i.e., $T = 0$), Eq.(A11) is reduced to Eq.(5.5.29) in the reference (Pedlosky, 1984).

### Data Availability Statement

The datasets used in this study are publicly available with the geoid undulation [$N(\lambda, \varphi)$] data at http://icgem.gfz-
potsdam.de/home, the density ($\rho$) data at https://www.nodc.noaa.gov/OC5/woa18/, and the annual mean wind stress ($\tau_\lambda, \tau_\varphi$

) data at http://iridl.ldeo.columbia.edu/SOURCES/.DASILVA/.SMD94/.climatology/.

### Author contributions





PCC discovered the problem, formulated the theory, calculated the two types of DOT, and prepared the manuscript.

**Competing interests**

The author declares that he has no conflict of interest.

**Acknowledgments**

Mr. Chenwu Fan's computational assistance is highly appreciated. The EIGEN-6C4 geoid undulation [$N(\lambda, \varphi)$] data was provided by the International Centre for Global Earth Models (ICGEM). The WOA18 annual mean temperature and salinity for the density $\rho$ data was obtained from the NOAA/NCEI. The SMD94 annual mean wind stress ($\tau_\lambda, \tau_\varphi$) data was archived from the International Research Institute for Climate and Society. The Research Office of the Naval Postgraduate School is also appreciated for paying the publication cost.

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



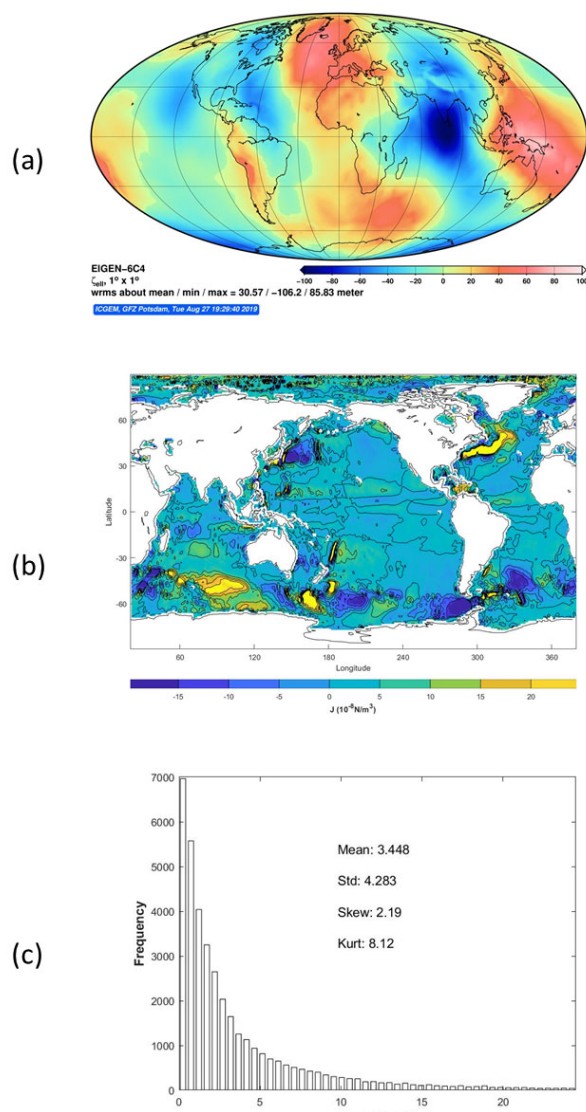

**Figure 1:** (a) Geoid undulation (N) from EIGEN-6C4 with 1°×1° resolution, computed from the website: http://icgem.gfz-potsdam.de/home, (b) contour plot of climatological annual mean GDF (unit: $10^{-8}$ Nm$^{-3}$) calculated using the NOAA/NCEI WOA18 annual mean temperature and salinity data and the EIGEN-6C4 geoid undulation (N) data, and (c) histogram of |GDF|. Note that GDF is simplified by 'J' here.




(a)

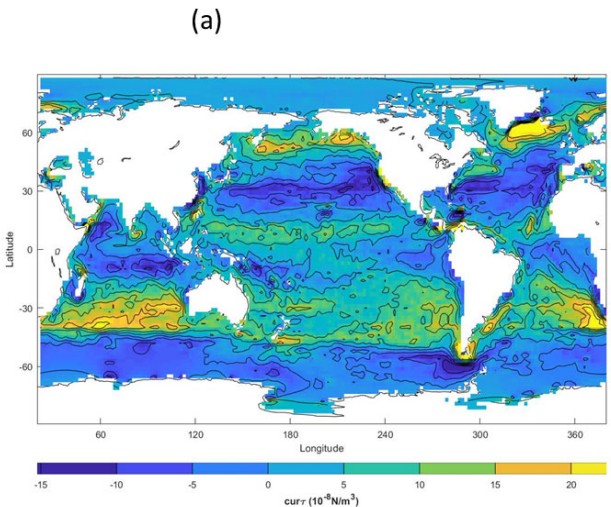

(b)

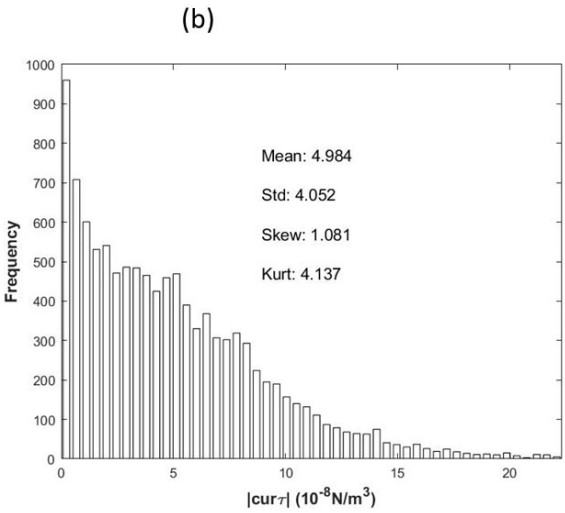


**Figure 2.** Climatological annual mean (curl τ) (unit: $10^{-8}$ Nm$^{-3}$) calculated using the COADS data: (a) contour plot of $(\mathrm{curl}\ \boldsymbol{\tau})$, and (b) histogram of $\left|\mathrm{curl}\ \boldsymbol{\tau}\right|$.



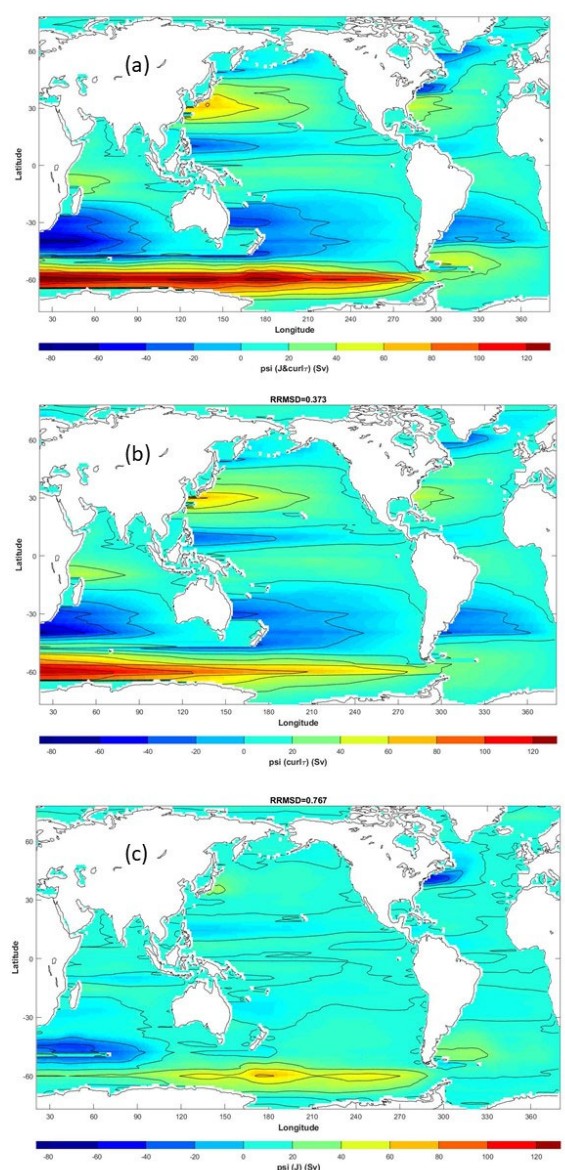

**Figure 3**. Sverdrup volume transport stream-function (unit: Sv, 1 Sv = $10^6$ m$^3$/s) with (a) both (curl $\tau$) and GDF, (b) (curl $\tau$), and (c) GDF.


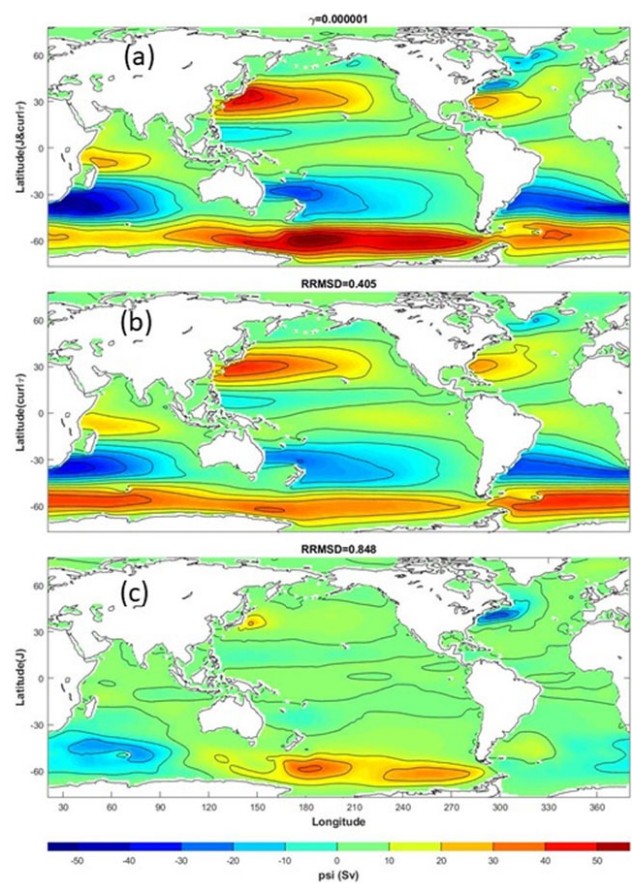


**Figure 4**. Stommel volume transport stream-function (unit: Sv, 1 Sv = $10^6$ m$^3$/s) with (a) both (curl τ) and GDF, (b) (curl τ), and (c) GDF.



**Table 1**. Three types of gravity and associated geopotential, marine geoid, dynamic ocean topography, and vertical.

| | True | Normal | Standard |
|---|---|---|---|
| Gravity | $\mathbf{g} = \mathbf{g}_n + \delta\mathbf{g}(\lambda, \varphi, z)$ | $\mathbf{g}_n = -g(\varphi)\mathbf{K}$ | $\mathbf{g}_s = -g_0\,\mathbf{k}$ |
| Geopotential | $\Phi = \Phi_n - T$ | $\Phi_n = g(\varphi)z$ | $\Phi_s = g_0 z$ |
| Geopotential Surfaces | Iso-Surfaces of Gravity Disturbance ($T$) | Ellipsoidal Surfaces | Spherical Surfaces |
| Marine Geoid | $z = N(\lambda, \varphi)$ | $z = N_n = 0$ | $z = N_s = 0$ |
| DOT Gradient | $\nabla D = \nabla[S - N(\lambda, \varphi)] \neq \nabla S$ | $\nabla^{(obl)} D = \nabla^{(obl)} S$ | $\nabla D = \nabla S$ |
| Vertical | In the direction of $\mathbf{g}$ Perpendicular to the $\Phi$-Surface | Perpendicular to the Earth Ellipsoidal Surface with $\mathbf{K}$ the unit vector (upward positive) | Perpendicular to the Earth Spherical Surface with $\mathbf{k}$ the unit vector (upward positive) |