# Peer review of "Gravity disturbance driven ocean circulation"

_Ocean Science, 2022_

## Author Comment (AC1)

Thank you very much for reviewing my manuscript.

**Major Responses**

First, your negative review is on the base of the concept that "*the only departures of a z-surface from an equipotential are due to the temporal variations in the gravitational field, i.e., the tidal forces*," which I disagree. The departure of a z-surface (i.e., spherical/ellipsoidal surface) from an equipotential (or more accurately say, the departure of an equipotential surface from a z-surface) represents the gravity disturbance, and is mainly caused by the nonuniform mass density inside the solid Earth rather than by the tidal forces. There are near 3 orders of magnitude difference between the two. You may find the definition of the gravity disturbance from the cited reference (Hackney & Featherstone, 2003) [or Equation (10) in the manuscript] and the quantification of the time-independent gravity disturbance by static gravity models with observations in geodetic community (see website http://icgem.gfz-potsdam.de/home). From a popular static gravity model EIGEN-6C4, the globally averaged absolute value of the gravity disturbance at $z = 0$ is estimated around 20 mGal (1 mGal $=10^{-5}$m s$^{-2}$). However, the sun and moon create time-dependent tidal forces that affect the measured value of gravity by about 0.3 mGal (from the Wikipedia website https://en.wikipedia.org/wiki/Gravity_anomaly). Such evidently large gravity disturbance (additional body force) exists no matter with or without tides and is totally neglected in ocean dynamics.

Second, your negative review is due to your interpretation of the geoid surface, "*I am pretty sure the community does not have in mind that following a z surface* (I think that you mean the geoid surface) *from boundary, to the centre, of the Indian Ocean requires one to climb roughly 100m against gravity,*" which I disagree. The geoid surface is an equipotential surface of the true gravity (normal gravity plus gravity disturbance). Following the geoid surface from boundary, to the center, of the Indian Ocean requires one to climb roughly 100m but NOT against the TRUE GRAVITY. Because the value of the true geopotential keeps the same on the geoid surface. You may find detailed information about the geoid in the Introduction Section of my manuscript or a geodesy textbook.

Third, your negative review is due to your treatment of the geoid surface as the sea surface elevation, "*I have not worked through the details, but related to the final point above is that the assumption of a rigid lid at z=0 in the derivation of the Sverdrup/Stommel/Munk equation (19) is unjustified given the order 100m variations in sea surface elevation in the authors coordinate system,*" which I disagree. The geoid surface and the sea surface elevation are independent and different. The geoid surface is an equipotential surface of the true gravity, and nothing to do with the sea surface elevation. The famous Brun's formula in geodesy shows that the gravity disturbance at $z = 0$ is the multiplication of the standard gravity (9.81 m/s$^2$) and the horizontal gradient of the geoid. See Equation (A1) of Sandwell and Walter (1997): Marine gravity anomaly from Geosat and ERS 1 satellite altimetry, JGR 102, 10,039-10,054 (see website: Marine gravity anomaly from Geosat and ERS 1 satellite altimetry (ucsd.edu)).

Fourth, your negative review is based on the statement, "*However, if a z surface is defined as an equipotential, then this term vanishes identically, calling into question the statements made in the abstract - the solution should not depend fundamentally on the choice of coordinate system,*" which I disagree. The z surface represents vertical location in the spherical (or local) coordinate system.

The equipotential surface is defined by the true gravity vector field. The z surface can never be defined as an equipotential surface. Even in geodesy, the gravity and equipotential are the major subjects. The equipotential is represented in the spherical coordinate system with z as the independent variable representing the vertical location.

**Responses to Your Assessment**

"*This paper makes the claim that the neglect of spatial variations in the Earth's gravity field due to the inhomogeneous composition of the Earth leads to substantial revisions to the classical solution for the Sverdrup/Stommel/Munk solution for the depth-integrated circulation of the oceans.*"

Yes. This manuscript shows that the gravity disturbance driven ocean circulation is comparable to the wind driven ocean circulation.

"*If true, this would certainly be a noteworthy result. However, having read the paper several times, and also the recent papers by the same author on the modifications to the equations of motion and oceanic/atmospheric Ekman layers due to the neglect of the same effect, I'm afraid that I am left scratching my head and wondering whether I'm missing something fundamental?*"

Yes. A fundamentally important forcing term, the gravity disturbance, is missing in ocean dynamics.

The Ekman layer dynamics and the Sverdrup/Stommel/Munk dynamics are related, but not the same. A paper published on the Ekman dynamics cannot be the reason to reject a paper on the Sverdrup/Stommel/Munk dynamics. I would like to know if any one has studied the effect of the gravity disturbance on the Sverdrup/Stommel/Munk dynamics, and if any paper has included the gravity disturbance in numerous publications on the Sverdrup/Stommel/Munk dynamics such as the Sverdrup transport.

**Response to Your Specific Comment**

"*I am also missing a good physical explanation in the paper of how the additional torques arise to drive the additional depth-integrated circulation? In equation (19), the additional source of vertical vorticity arises through the projection of the baroclinic production of vortictiy onto the vertical component of the vorticity equation.*"

Eq(19) is the curl of depth-integrated circulation from Eq.(18), i.e., the horizontal momentum equation for steady-state low Rossby number with friction. Eq(18) shows the balance among the Coriolis force, horizontal pressure gradient force, friction, and the additional gravity disturbance. Due to the Boussinesq approximation, all the terms are independent on the density or density anomaly (i.e., density minus constant reference density) except the gravity disturbance term, which is the gravity disturbance multiplied by the density. That is the physical reason of the additional source of the vertical vorticity arises through the projection of the baroclinic production of vortictiy onto the vertical component of the vorticity equation, i.e., the vertically integrated Jacobian of the density and gravity disturbance.

**Response to the Recommendation**

"*So, in summary, I'm afraid that I cannot recommend this manuscript for publication as I feel that the results rely on a particular and, in my honest opinion, rather odd choice of coordinate system. If I have missed something fundamental, then I apologise in advance and am happy to stand corrected.*"

I hope you may change your recommendation after reading my responses. Also, I can't get my head around your critics on the choice of coordinate system. I just use the common z coordinate system like most of oceanographers do.

"*Finally, I note that there is quite a lot of overlapl between this and the author's three previous papers on a similar topic, especially in the preliminary material. If the manuscript is published, then I would suggest pruning the material down to focus on that which is novel to this contribution.*"

I will revise my manuscript before publication to reduce redundancy.

---

## Author Comment (AC3)

**Normal (or called Effective) Gravity**
**(Uniform Mass Density inside the Solid Earth)**
**Normal Geopotential Surface = z Surface**

[Figure]

The two figures are from the website:

https://atoc.colorado.edu/~cassano/atoc5050/Lecture_Notes/hh_ch1.pdf

$z = 0$

$-g(\varphi)$ **K**     Normal Vertical

z = constant

Normal Horizontal

Normal Geopotential Surface

Any movement on z = constant (i.e., z surfaces)
is not against the effective gravity.
The normal geopotential surface coincides
with the z-surface.

**True Gravity**
**(Non-uniform Mass Density inside the Solid Earth): $g(\lambda, \varphi, z) = - g(\varphi) \mathbf{K} + \delta\mathbf{g}$**
**$\delta\mathbf{g}$ = Gravity Disturbance**

Equipotential Corresponding to true gravity Is the True Geopotential

Extra mass

Less mass

EARTH

Geoid ($z = N$) is an equipotential surface at top ocean

[Figure]

EIGEN−6C4
$\zeta_{ell}, 1° \times 1°$
wrms about mean / min / max = 30.57 / −106.2 / 85.83 meter
ICGEM, GFZ Potsdam, Tue Aug 27 19:29:40 2019

(Data from EIGEN-6C4)

$85.83$ m $> N(\lambda,\varphi) > -106.20$ m

$\mathbf{K}$

$z = 0$

Geoid ($z = N$)

True Geopotential Surface (Horizontal)

$g(\lambda, \varphi, z)$ True Vertical

$g(\lambda, \varphi, z)$ True Vertical

$z =$ const

(1) Any movement along the geoid surface (true horizontal surface), z= N($\lambda,\varphi$), (-106.20 m to 85.83 m, from EIGEN-6C4) is not against the true gravity.

(2) Any movement on the z-surface is against the true gravity. An additional force, Gravity Disturbance, shows up in the z-surface momentum equations.

---

## Author Comment (AC4)

**Response to the Simple Thought Experiment**

 " *I ask the author to consider the following situation where the planet is an aqua planet, and the ocean is not in motion.  This requires that in situ density is constant at each point on the real geoid surface (not the ellipsoidal approximation to it).  The author's GFD is however non-zero and large in this situation; that is, his equation (22). But this turns out only to be that he has not chosen his vertical distance to be measured from the real geopotential.  Rather he has chosen the zero of his height to be in an ellipsoidal surface.  So his equations show substantial motion, but we know that there should be no motion.*"

I disagree.

Equation (22) in the manuscript is written by

$$\text{GDF} = \int_{-H}^{0} \left( \frac{\partial \rho}{\partial x} \frac{\partial T}{\partial y} - \frac{\partial \rho}{\partial y} \frac{\partial T}{\partial x} \right) dz = \int_{-H}^{0} \left[ \mathbf{k} \bullet \left( \nabla \rho \times \nabla T \right) \right] dz$$

Consider that the in-situ density is constant at each point on the true geopotential  surface, i.e.,  the isopycnal surface coincides with the true geopotential surface. This requires that the two vectors $\nabla \rho$ and $\nabla T$ are parallel, i.e.,

$$\nabla \rho \times \nabla T = 0$$

which leads to

$$\text{GDF} = 0$$

which shows that the GDF does not drive any motion. It is the opposite outcome as you thought. This simple thought experiment demonstrates the merit of the manuscript.

**Response on the Geopotential and Geopotential Surface**

"*The development of the equations with respect to the geoid is done in textbooks, for example in the early pages of the text "Fundamentals of Ocean Climate Models" by S. M. Griffies, published in 2004.  These ocean models do not put the ocean in motion if the in situ density is constant on geopotential surfaces.*"

The geopotential and geopotential surface used in oceanography and meteorology including in the text "Fundamentals of Ocean Climate Models" by S. M. Griffies are the normal geopotential and normal geopotential surface, but not the  TRUE GEOPOTENTIAL and TRUE GEOPOTENTIAL SURFACE.

The two attached figures illustrate the difference between the normal gravity which is called the effective gravity and used in oceanography and meteorology, and the true gravity which is the most important variable in geodesy.

Figure A shows the main features of the effective gravity [-g(φ)**K**]: (1) it is determined from the solid Earth with rotation and uniform mass density; (2) the unit vector **K** is perpendicular to the z surface (z = constant) and points the normal vertical; (3) the z surface is the normal horizontal and coincides with the normal geopotential surface; (4) any movement on the z surface (i.e., normal geopotential surface) is not against the normal gravity.

**(A) Normal (or called Effective) Gravity**
**(Uniform Mass Density inside the Solid Earth)**
**Normal Geopotential Surface = z Surface**

[Figure]

The two figures are from the website:
https://atoc.colorado.edu/~cassano/atoc5050/Lecture_Notes/hh_ch1.pdf

[Figure]

[Figure]

Any movement on z = constant (i.e., z surfaces)
is not against the effective gravity.
The normal geopotential surface coincides
with the z-surface.

**Fig. A. Illustration of normal gravity, normal geopotential, normal vertical, and normal horizontal, which are used in atmospheric and oceanic dynamics.**

Figure B shows the main features of the true gravity [$\mathbf{g}(\lambda, \varphi, z) = -g(\varphi)\mathbf{K} + \delta\mathbf{g}$ ]: (1) it is determined from the solid Earth with rotation and non-uniform mass density; (2) the true gravity has never been used in oceanography and meteorology; (3) the true gravity vector $\mathbf{g}(\lambda, \varphi, z)$ is perpendicular to the true geopotential surface such as the geoid surface, which represents the true horizontal; (4) any movement on the true geopotential surface is not against the true gravity; (5) any movement on the z-surface is against the true gravity. An additional force, the gravity disturbance (T), shows up in the z-surface momentum equations, such as in Equation (18) of the manuscript.

[Figure]

**Fig. B. Illustration of true gravity, true geopotential, true vertical, and true horizontal, which should be used in atmospheric and oceanic dynamics**

---

## Author Comment (AC5)

**Response on the Geopotential and Geopotential Surface**

*"I have read the manuscript by Peter Chu, and while I found it quite thought-provoking, I am forced to conclude that it is actually quite misleading and do not recommend publication in its present form. It seems to me that the mistake the author is making is to formulate the equations of motion in spherical coordinates from the beginning. This is not my understanding of how the equations of motion used by models of the atmosphere and ocean are formulated. Rather, these models use a coordinate system in which the vertical direction is defined as being perpendicular to geopotential surfaces so that gravity always points along the vertical direction with no horizontal component."*

The geopotential and geopotential surface used in oceanography and meteorology are the normal geopotential and normal geopotential surface, but not the  TRUE GEOPOTENTIAL and TRUE GEOPOTENTIAL SURFACE.

The two attached figures illustrate the difference between the normal gravity which is called the effective gravity and used in oceanography and meteorology, and the true gravity which is the most important variable in geodesy.

Figure A shows the main features of the effective gravity $[-g(\varphi)\mathbf{K}]$: (1) it is determined from the solid Earth with rotation and uniform mass density; (2) the unit vector $\mathbf{K}$ is perpendicular to the z surface (z =  constant) and  points the normal vertical;  (3) the z surface is the normal horizontal and coincides with the normal geopotential surface; (4)  any movement on the z surface (i.e., normal geopotential surface) is not against the normal gravity.

**(A) Normal (or called Effective) Gravity**
**(Uniform Mass Density inside the Solid Earth)**
**Normal Geopotential Surface = z Surface**

[Figure]

The two figures are from the website:
https://atoc.colorado.edu/~cassano/atoc5050/Lecture_Notes/hh_ch1.pdf

$-g(\varphi)$ **K**    Normal Vertical

z = constant
Normal Horizontal
Normal Geopotential Surface

Any movement on z = constant (i.e., z surfaces)
is not against the effective gravity.
The normal geopotential surface coincides
with the z-surface.

**Fig. A. Illustration of normal gravity, normal geopotential, normal vertical, and normal horizontal, which are used in atmospheric and oceanic dynamics.**

Figure B shows the main features of the true gravity $[\mathbf{g}(\lambda, \varphi, z) = -g(\varphi)\mathbf{K} + \delta\mathbf{g}\ ]$: (1) it is determined from the solid Earth with rotation and non-uniform mass density; (2) the true gravity has never been used in oceanography and meteorology; (3) the true gravity vector $\mathbf{g}(\lambda, \varphi, z)$ is perpendicular to the true geopotential surface such as the geoid surface, which represents the true horizontal; (4) any movement on the true geopotential surface is not against the true gravity; (5) any movement on the z-surface is against the true gravity. An additional force, the gravity disturbance ($T$), shows up in the z-surface momentum equations, such as in Equation (18) of the manuscript.

[Figure]

**Fig. B. Illustration of true gravity, true geopotential, true vertical, and true horizontal, which should be used in atmospheric and oceanic dynamics.**

**Response to the Coordinate System**

The ocean dynamics to include the effect of the gravity disturbance shouldn't be depend on coordinate system. I use the vector form to redrive the combined Sverdrup-Stommel-Munk equation starting from Equation (12) in the manuscript.

**2 Basic equations**

Large-scale ocean circulation under the Boussinesq approximation is governed by the momentum equation (Chu, 2021a)

$$\rho_0 \left[ \frac{D\mathbf{U}}{Dt} + f\mathbf{k} \times \mathbf{U} \right] = -\nabla_3 p + \rho \mathbf{g} + \rho_0 (\mathbf{F}_h + \mathbf{F}_v) \qquad (12)$$

and the continuity equation

$$\nabla \bullet \mathbf{U} + \frac{\partial w}{\partial z} = 0 \qquad (13)$$

where $\nabla_3 \equiv \nabla + \mathbf{k}\partial / \partial z$, is the three-dimensional vector differential operator; $D/Dt$ is total time derivative; $\mathbf{U} = (u, v)$, is the 2D velocity vector in $(\mathbf{i}, \mathbf{j})$ surfaces; $w$ is the z-component velocity; $p$ is the pressure; $\rho$ is the sea water density; $\rho_0 = 1{,}028$ kg m$^{-3}$, is the reference density; $f = 2\Omega \sin \varphi$, is the Coriolis parameter with $\Omega = 2\pi/(86164$ s$)$; $(\mathbf{F}_h, \mathbf{F}_v)$ are the frictional forces with lateral and z-directional shears represented by

$$\mathbf{F}_h = A\nabla^2 \mathbf{U}, \quad \mathbf{F}_v = \frac{\partial}{\partial z}\left( K \frac{\partial \mathbf{U}}{\partial z} \right) \qquad (14)$$

where $(A, K)$ are the corresponding eddy viscosities.

With the constant reference density $\rho_0$, we can define a reference hydrostatic pressure $p_0$ that exactly balance the component of the true gravity $\mathbf{g}$ $[= \nabla_3 \Phi, \quad \Phi = g_0 z - T \approx g_0 z - g_0 N(\lambda, \varphi)]$ associated with reference density; i.e.,

$$-\nabla_3 p_0 - \rho_0 \nabla_3 \Phi = 0 \qquad (15)$$

where

$$p_0 = -\rho_0 g_0 z + \rho_0 T \approx -\rho_0 g_0 (z - N) \qquad (16)$$

is the hydrostatic pressure. Subtraction of (15) from (12) leads to

$$\rho_0 \left[ \frac{D\mathbf{U}}{Dt} + f\mathbf{k} \times \mathbf{U} \right] = -\nabla \hat{p} + (\rho - \rho_0)\nabla T + \rho_0 (\mathbf{F}_h + \mathbf{F}_v) \qquad (17a)$$

$$\frac{\partial \hat{p}}{\partial z} = -(\rho - \rho_0) g_0 \qquad (17b)$$

where $\hat{p} = p - p_0$, is the dynamic pressure.

**3 Combined Sverdrup-Stommel-Munk equation**

For steady-state low Rossby number (negligible nonlinear advection) flow with friction (i.e., $D\mathbf{U}/Dt = 0$, and $\mathbf{F}_h \neq 0$, $\mathbf{F}_v \neq 0$), Eq.(17a) is simplified into

$$\rho_0 \left[ f\mathbf{k} \times \mathbf{U} - A\nabla^2\mathbf{U} - \frac{\partial}{\partial z}\left( K\frac{\partial \mathbf{U}}{\partial z} \right) \right] = -\nabla\hat{p} + (\rho - \rho_0)\nabla T \tag{18}$$

where (14) is used for $\mathbf{F}_h$ and $\mathbf{F}_v$. The turbulent momentum flux is given by

$$\rho_0 K \frac{\partial \mathbf{U}}{\partial z}\Big|_{z=0} = \boldsymbol{\tau} \tag{19a}$$

at the rigid-lid ocean surface ($z = 0$) with $\boldsymbol{\tau}$ the wind stress, and given by

$$K\frac{\partial \mathbf{U}}{\partial z}\Big|_{z=-H} = \gamma\mathbf{M}, \quad \mathbf{M} \equiv \int_{-H}^{0} \mathbf{U}dz \tag{19b}$$

at the lower boundary ($z = -H$). Here, $\mathbf{M}$ is the volume transport; $\gamma$ is the Rayleigh friction coefficient (Stommel 1948). When $\gamma = 0$, Eq.(19b) represents negligible turbulent momentum flux at $z = -H$ (Sverdrup 1947, Munk 1950). Integration of (18) in the z-direction from $z = -H$ to $z = 0$ and use of (19a) and (19b) leads to

$$\left[ f\mathbf{k} \times \mathbf{M} - A\nabla^2\mathbf{M} - (\boldsymbol{\tau} - \gamma\mathbf{M}) \right] = -\int_{-H}^{0}\nabla\hat{p}dz + \int_{-H}^{0}\left[ (\rho - \rho_0)\nabla T \right]dz \tag{20}$$

Curl of the vector equation (20) gives

$$\nabla \times \left[ f\mathbf{k} \times \mathbf{M} - A\nabla^2\mathbf{M} - (\boldsymbol{\tau} - \gamma\mathbf{M}) \right] = \int_{-H}^{0}\left[ \nabla\rho \times \nabla T \right]dz \tag{21}$$

Let the volume transport stream-function ($\Psi$) be defined by

$$\nabla\Psi = -\frac{1}{\rho_0}\mathbf{k} \times \mathbf{M}, \quad \text{i.e.,} \quad \mathbf{M} = \rho_0\mathbf{k} \times \nabla\Psi \tag{22}$$

Substitution of (22) into (21) leads to

$$\nabla \times \left[ -f\nabla\Psi - A\nabla^2(\mathbf{k} \times \nabla\Psi) + \gamma(\mathbf{k} \times \nabla\Psi) \right] = \frac{1}{\rho_0}\left[ \nabla \times \boldsymbol{\tau} + \int_{-H}^{0}(\nabla\rho \times \nabla T)dz \right] \tag{23}$$

Since

$$\nabla \times \left(\mathbf{k} \times \nabla \Psi\right) = \mathbf{k}\nabla^{2}\Psi, \ \nabla \times \left(-f\nabla \Psi\right) = \beta\frac{\partial \Psi}{\partial x}, \ \ \beta \equiv \frac{df}{dy} \ \ (\beta \text{ coefficient}) \tag{24}$$

substituting (24) into (23) and conducting inner product with the unit vector $\mathbf{k}$, we obtain the combined Sverdrup-Stommel-Munk equation

$$-A\nabla^{4}\Psi + \gamma\nabla^{2}\Psi + \beta\frac{\partial \Psi}{\partial x} = \frac{1}{\rho_{0}}\left[\operatorname{curl}\boldsymbol{\tau} + \int_{-H}^{0}\mathbf{k}\bullet\left(\nabla\rho\times\nabla T\right)dz\right] \tag{25}$$

with an additional gravity disturbance forcing (GDF),

$$\text{GDF} = \int_{-H}^{0}\mathbf{k}\bullet\left(\nabla\rho\times\nabla T\right)dz = \int_{-H}^{0}J(\rho,T)dz, \ \ J(\rho,T) \equiv \frac{\partial \rho}{\partial x}\frac{\partial T}{\partial y} - \frac{\partial T}{\partial x}\frac{\partial \rho}{\partial y} \tag{26}$$

where $J(\rho, T)$ is the Jacobian of $\rho$ and $T$.